# From Personalized to Precision Medicine in Oncology: A Model-Based Dosing Approach to Optimize Achievement of Imatinib Target Exposure

**DOI:** 10.3390/pharmaceutics15041081

**Published:** 2023-03-28

**Authors:** Sylvain Goutelle, Monia Guidi, Verena Gotta, Chantal Csajka, Thierry Buclin, Nicolas Widmer

**Affiliations:** 1Service de Pharmacie, GH Nord, Hospices Civils de Lyon, 69002 Lyon, France; 2Univ. Lyon, Université Claude Bernard Lyon 1, UMR CNRS 5558, LBBE—Laboratoire de Biométrie et Biologie Évolutive, 69100 Villeurbanne, France; 3Univ. Lyon, Université Claude Bernard Lyon 1, ISPB—Faculté de Pharmacie de Lyon, 69008 Lyon, France; 4Service of Clinical Pharmacology, Lausanne University Hospital and University of Lausanne, 1011 Lausanne, Switzerland; monia.guidi@chuv.ch (M.G.); nicolas.widmer@chuv.ch (N.W.); 5Center for Research and Innovation in Clinical Pharmaceutical Sciences, University Hospital and University of Lausanne, 1011 Lausanne, Switzerland; 6Institute of Pharmaceutical Sciences of Western Switzerland, University of Geneva and University of Lausanne, 1211 Geneva, Switzerland; 7Pediatric Pharmacology and Pharmacometrics, University of Basel Children’s Hospital, 4056 Basel, Switzerland; 8School of Pharmaceutical Sciences, University of Geneva, 1205 Geneva, Switzerland; 9Pharmacy of the Eastern Vaud Hospitals, 1847 Rennaz, Switzerland

**Keywords:** imatinib, pharmacokinetics, model-informed precision dosing, oncology, antineoplasic agents

## Abstract

Imatinib is a targeted cancer therapy that has significantly improved the care of patients with chronic myeloid leukemia (CML) and gastrointestinal stromal tumor (GIST). However, it has been shown that the recommended dosages of imatinib are associated with trough plasma concentration (Cmin) lower than the target value in many patients. The aims of this study were to design a novel model-based dosing approach for imatinib and to compare the performance of this method with that of other dosing methods. Three target interval dosing (TID) methods were developed based on a previously published PK model to optimize the achievement of a target Cmin interval or minimize underexposure. We compared the performance of those methods to that of traditional model-based target concentration dosing (TCD) as well as fixed-dose regimen using simulated patients (*n* = 800) as well as real patients’ data (*n* = 85). Both TID and TCD model-based approaches were effective with about 65% of Cmin achieving the target imatinib Cmin interval of 1000–2000 ng/mL in 800 simulated patients and more than 75% using real data. The TID approach could also minimize underexposure. The standard 400 mg/24 h dosage of imatinib was associated with only 29% and 16.5% of target attainment in simulated and real conditions, respectively. Some other fixed-dose regimens performed better but could not minimize over- or underexposure. Model-based, goal-oriented methods can improve initial dosing of imatinib. Combined with subsequent TDM, these approaches are a rational basis for precision dosing of imatinib and other drugs with exposure–response relationships in oncology.

## 1. Introduction

Imatinib is a selective and potent tyrosine kinases inhibitor approved notably for the treatment of chronic myeloid leukemia (CML) and gastrointestinal stromal tumor (GIST). It has been a major breakthrough in cancer therapy [1], as it specifically inhibits the genetically altered molecular targets (i.e., tyrosine kinase proteins) involved in the pathogenesis of both those two conditions and has been associated with significant increase in response rates [2,3]. Owing to its innovative mechanism of action and its clinical success, imatinib has been considered as a prototypical example of targeted drug therapy [4] and a successful application of the concept of personalized medicine [5]. Since imatinib’s discovery, significant advances in this area have been made, especially in oncology [6], although the actual public health benefit expected from a generalization of this approach is still questionable [7].

In 2015, the precision medicine initiative was launched in the USA [8]. Precision medicine includes the concepts of personalized medicine (i.e., reliance on genetic or acquired biomarkers for therapeutic decisions), but it is also a step beyond as it encompasses identification, quantification and clinical applications of all potential determinants of individual response in prevention and treatment strategies. The initiative also promotes the utilization of big data in health as well as the deployment of computational tools to analyze them.

In his allocution in 2015, President Obama also stressed the importance of “figuring out the right dose of medicine” in personalized medicine (https://obamawhitehouse.archives.gov/node/333101, accessed on 12 December 2022). In line with other researchers [9,10], we believe that precision dosage design of drugs has been overlooked in personalized medicine so far, and that major progress can be made in this area. While selecting the right drug for a given individual based on his/her characteristics is important, prescribing the best dose of such drug also matters to optimize drug response. Precision dosing notably includes drug dosing based on models to better consider PK-PD variability and better achieve target objectives in a goal-oriented approach. This can be performed both a priori (without any concentration measurement) and a posteriori, with a further readjustment of the dosing schedule based on the patient’s actual exposure. In this paper, we aim at showing how a priori precision dosing can be applied using imatinib as an illustrative example.

The approved dosages of imatinib in adults are 400 mg once daily in GIST and chronic phase of CML, and 600 mg in CML accelerated phase and blast crisis. The dosage may be increased up to a maximum of 800 mg per day in patients with CML. The recommended dosage regimen of imatinib in chronic phase CML, 400 mg/day, has been based on in vitro activity and a mathematical model of hematological dose–response indicating that 400 mg was sufficient to provide optimal response in most patients [1,11]. Since then, it has been shown that a higher dose may be necessary to optimize cytogenetic response and survival [12]. Indeed, a higher dosage of imatinib has been associated with earlier and higher rates of cytogenetic and molecular response in patients with CML [13,14,15].

Since imatinib’s approval in the early 2000s, there has been mounting evidence of concentration–response relationships for both CML and GIST therapy. In patients with CML, several observational studies reported that achieving an imatinib trough concentration (Cmin) above a target of about 1000 ng/mL was associated with improved rates of hematological and molecular response [16,17,18]. In patients with GIST, it has been shown that Cmin values above 1100 mg/L were associated with significantly longer time to disease progression [19]. Although some adverse reactions were more frequent in patients with the highest Cmin, in the study from Larson and colleagues [16], data supporting an upper bound for imatinib concentration are limited [20].

Because of this concentration-dependent activity and large interindividual PK variability [16], it has been suggested that patients could benefit from therapeutic drug monitoring of imatinib (TDM), and real-life data appear to confirm the clinical value of TDM [20,21,22,23,24,25]. However, TDM services may not be available for all patients treated with imatinib. In addition, because of the relatively long half-life of imatinib (mean value of 19 h [11]), it may take many days to adequately measure concentrations at the steady-state, then adjust dosage and control the achievement of target Cmin. As a result, slow response to imatinib may occur because of initial suboptimal exposure.

The objectives of the present study were to design a new model-based approach for optimal targeting of imatinib Cmin and evaluate the ability of this novel approach to achieve a target Cmin interval compared with fixed-dose regimens and another model-based approach.

## 2. Materials and Methods

### 2.1. Pharmacokinetic Model

The model published by Widmer et al. [26] was used to build the new dosing method. Briefly, this model was based on the nonlinear mixed-effects population analysis of 321 plasma concentration of imatinib measured in 59 patients with CML or GIST. The final model was a one-compartment model with linear absorption and elimination. The absorption rate constant Ka was a non-random parameter, with an estimated value of 0.609 h^−1^ in all individuals. Population median and coefficient of variation of apparent oral clearance (CL/F) and volume of distribution (V/F) were 14.3 L/h (36%) and 347 L (63%), respectively. These two parameters were assumed to have a lognormal distribution. Significant covariance was found between CL/F and V/F, with an estimated correlation coefficient of 0.8. Because imatinib oral bioavailability is close to 100% [11], we assumed that bioavailability was 100%, so CL/F and V/F will be simply denoted as *CL* and *V* in the rest of the manuscript.

Several covariates were found to influence imatinib *CL* and *V*. The relationships between typical values (*TV*) of *CL* and *V* and covariates were modeled as follows:(1)$${TV}_{CL}=14.3+5.42\cdot \frac{BW-70}{70}+1.49\cdot male-1.49\left(1-male\right)-5.81\cdot \frac{AGE-50}{50}-0.806\cdot path+0.806\cdot (1-path)$$
(2)$${TV}_{V}=347+46.2\cdot male-46.2\cdot \left(1-male\right)$$
where *TV_CL_* is the typical value of *CL*, *TV_V_* is the typical value of *V*, *BW* is patient’s body weight (in kg), *AGE* is the patient’s age (in years). Male is a binary variable indicating male gender (1 if male, 0 if female), and path is a binary variable indicating the disease (1 if GIST, 0 if CML). In summary, a priori imatinib clearance increases with increasing *BW*, decreases with increasing age, and is greater in males and patients with CML compared with females and GIST patients, respectively. Typical imatinib *V* is greater in male than in female subjects.

The performance of this model in predicting imatinib concentration collected during routine TDM was assessed in a separate study, with two datasets (65 and 20 subjects) not used for model building. The results showed good overall predictive performance of the model estimates based on a Bayesian Maximum A Posteriori (MAP) calculation of individual PK parameters [27].

### 2.2. Model Discretization

Our approach for optimal targeting of a concentration interval has been described in detail in a previous publication [28]. It is inspired by previous works from D’Argenio and colleagues who used a discrete prior distribution of PK parameters to optimize model-based control of drug exposure [29,30].

The lognormal distribution of individual *CL* and *Vd* from Widmer’s model can be written as follows:(3)$${CL}_{i}={TV}_{CL}\times exp(\eta_{CLi})$$
and
(4)$$V_{i}={TV}_{V}\times exp(\eta_{Vi})$$
where *TV_CL_* and *TV_V_* are as defined in Equations (1) and (2), and *η_CLi_* and *η_Vi_* are random variables that follow a normal distribution with zero mean and variance–covariance matrix *Ω*^2^ defined as follows:(5)$$\Omega^{2}=\left[\begin{array}{cc} \omega^{2}(CL) & \omega^{2}(CL,V)\\ \omega^{2}(CL,V) & \omega^{2}(V) \end{array}\right]=\left[\begin{array}{cc} 0.127 & 0.179\\ 0.179 & 0.396 \end{array}\right]$$

The joint distribution of η parameters was discretized into a grid of 81 discrete values. Those 81 values were obtained by random sampling from the corresponding bivariate normal distribution, using the multivariate normal random numbers routine of Matlab (version 2018b, Mathworks, Natick, MA, USA), and their probabilities were obtained from the corresponding probability function values. The choice of 81 as the number of grid points was inspired by a previous work from Katz and D’Argenio which showed that this number was sufficient to obtain a correct approximation of a bivariate continuous distribution [31]. Indeed, this number may be considered large, considering that a nonparametric population model of amikacin developed in 634 patients yielded a grid of 76 support points [32]. The median and variance of the discrete distribution were −0.024 (0.114) and −0.057 (0.390) for *η_CLi_* and *η_Vi_*, respectively, with a covariance of 0.167, very similar to values of the original continuous distribution. The prior value of Ka was fixed to 0.609, in accordance with the model from Widmer et al. [26].

This grid of 81 values of random effects allowed deriving a discrete prior distribution of *CLi* and *Vi* for each patient. Indeed, Equations (1) and (2) can be used to calculate typical values of *CL* and *V* based on the patients’ covariates and discrete prior distributions of *CL* and *V* can be obtained by incorporating the vectors of 81 values of random effects (*ηCLi* and *ηVi*) into Equations (3) and (4).

An example of such bivariate prior distribution of *CLi* and *Vi* is shown in Figure 1.

The nonparametric, discrete PK model can be used to calculate a priori Cmin distribution for various imatinib dosage regimens. Let us take the example of a virtual patient, a 60-year-old woman weighing 60 kg, with CML. Based on Equations (1) and (2), her median a priori values of *CL* and *V* are 11.7 L/h and 300.8 L, respectively. However, using the discrete distribution, there is an a priori distribution of 81 values of *CL* and *V* similar to that shown in Figure 1.

Then, one can simulate the PK profile of imatinib under a given dosage regimen in this patient. The trough concentration of imatinib after multiple oral doses at the steady-state (*Css_min_*, further denoted as *Cmin* for ease of reading) is given by the following equation:(6)$${Css}_{min}=\frac{F.Dose.Ka}{V.(Ka-Ke)}\cdot \left(\frac{1}{1-e^{-Ke.tau}}-\frac{1}{1-e^{-Ka.tau}}\right)$$
where *F* is the oral bioavailability (here assumed to be equal to 1, as previously mentioned), Dose is the dose (in mg), *Ka* and *V* are as defined above, *tau* is the dosing interval (in hours) and *Ke* is the elimination rate constant, which is equal to *CL*/*V*.

The discrete distribution of *CL* and *V* parameters results in a discrete a priori distribution of Cmin for a given dosage regimen in this patient. Such a distribution is shown in Figure 2 for two dosage regimens, 400 mg/24 h and 600 mg/24 h.

### 2.3. The Target Interval Dosing (TID) Approach

For a patient who received a given dosage regimen, there are *n* (here *n* = 81) possible Cmin values, corresponding to the *n* vectors of PK parameters (*CL_i_*, *Vd_i_*) in the discrete collection, and each *Cmin_i_* has an a priori probability, *prob_i_*. The sum of probi is equal to one.

If the target interval of *Cmin* is defined by a lower bound *L* and an upper bound *U*, the a priori probability that *Cmin* is within the interval [*L*, *U*] for a given dosage regimen is given by
(7)$$\sum_{i=0}^{n}{prob}_{i}({Cmin}_{i}\in \left[L;U\right])$$

The optimal dosage (*ODint*) for achieving the target Cmin interval [*L*, *U*] is the combination of dose and dosing interval that maximizes the a priori probability as defined by Equation (7). Mathematically, this translates into
(8)$${OD}_{int}\left(Dose,tau\right)=arg\left\{\underset{\mathrm{Dose},\mathrm{tau}}{\max}\left[7\right]\right\}$$

In the case of imatinib, we evaluated 14 possible initial dosage regimens of increasing intensity, with daily doses ranging from 100 to 800 mg with a 100 mg increment, and 3 dosing intervals, q8h, q12h and q24h. These 14 regimens were: 100 mg/24 h, 200 mg/24 h, 100 mg/12 h, 300 mg/24 h, 100 mg/8 h, 400 mg/24 h, 200 mg/12 h, 500 mg/24 h, 600 mg/24 h, 300 mg/12 h, 200 mg/8 h, 700 mg/24 h, 800 mg/24 h and 400 mg/12 h. The daily dose was limited to 800 mg, which is the maximum recommended daily dose for imatinib.

Instead of targeting an interval of concentration, one may be interested in minimizing the a priori probability of underexposure or overexposure. For imatinib, it is relevant to minimize underexposure, i.e., to avoid low *Cmin*. In such case, the probability of interest is
(9)$$\sum_{i=0}^{n}{prob}_{i}({Cmin}_{i}<L)$$

The optimal dosage regimen (*OD_Lmin_*) is the one that minimizes this probability:(10)$${OD}_{Lmin}\left(Dose,tau\right)=arg\left\{\underset{\mathrm{Dose},\mathrm{tau}}{\min}\left[9\right]\right\}$$

Because a very high dose may be necessary to achieve a zero probability of underdosing, it may be more clinically relevant to identify the dosage associated with a specified low but non-zero probability of underdosing, for example 5%:(11)$${OD}_{L5\%}\left(Dose,tau\right)=arg\left\{{\left[9\right]}_{\mathrm{Dose},\mathrm{tau}}<0.05\right\}$$

*OD_int_*, *ODL_min_* and *OD_L_*_5%_ are three ways to perform a general dosing approach that we have called target interval dosing (TID).

Figure 3 summarizes the various steps of the TID approach for imatinib dosing.

### 2.4. Dosing Based on the Traditional Model-Based Approach

The traditional model-based dosing approach uses a single vector of PK parameters, the typical values provided by Equations (1) and (2) as a priori values and considers a single concentration value as the target. This approach will be denoted as target concentration dosing (TCD), in reference to Target Concentration Intervention (TCI) which has been used to qualify the activity consisting in model-based dose adjustment based on TDM results [33]. Most of the time, the target is fixed at the midpoint of the therapeutic interval. In our example, the dose targeting the center of the Cmin interval can be calculated by rearranging Equation (3) as follows (it is assumed that F = 1):(12)$${Dose}^{target}=\frac{C_{min}^{target}.Vd.(Ka-Ke)}{Ka}\cdot \frac{1}{\frac{1}{1-e^{-Ke.tau}}-\frac{1}{1-e^{-Ka.tau}}}$$

### 2.5. Evaluation of TID and Other Dosing Methods in Simulated Patients

Virtual populations of patients, defined by PK parameters and covariate values of the imatinib model were created to assess performance the TID and alternative dosing methods.

We considered four groups for categorical covariates: females with CML, females with GIST, males with CML and males with GIST.

For each group, 200 values of body weight and age were randomly sampled from two independent truncated normal distributions, with mean ± standard deviation (minimum–maximum) values of 64 ± 11 kg (40–110) in females and 79 ± 11 kg (40–110) in males for body weight, and 62 ± 13 years (20–90) for age. These values were representative of individual data from 85 patients with GIST treated with imatinib [27]. Then, typical values of *CL* and *V* were calculated for each patient using Equations (1) and (2). True simulated parameter values were calculated using Equations (3) and (4), based on calculated typical values and random effect (*η_CLi_* and *η_Vi_*) values obtained by random sampling from the bivariate joint distribution.

We evaluated the ability of the TID and TCD dosing methods, as well as standard fixed-dose approaches to achieve a common imatinib target interval of 1000–2000 ng/mL in the 4 simulated populations defined above.

For the TID method, three optimal dosages (OD) were derived for each target interval:-*ODint*, the dosage maximizing the attainment of the target interval as defined in Equation (8);-*ODL_min_*, the dosage minimizing underexposure (Cmin < *L*, see Equation (10));-*ODL*_5%_, the dosage associated with a priori probability of underexposure less than 5% (Equation (11)).

For the traditional model-based dosing approach (TCD), we used the mid-interval as the target *Cmin*, i.e., 1500 mg/L. The dose was calculated using Equation (12) for both 12 h and 24 h dosing intervals, rounded to the nearest hundred, and capped to a maximum of 800 mg and 400 mg for once and twice daily dosing, respectively.

We also evaluated four fixed-dose regimens in each simulated patient: 400 mg/24 h, 600 mg/24 h, 200 mg/12 h and 300 mg/12 h. Once daily dosages of 400 mg and 600 mg are dosages mentioned in imatinib’s European summary of product characteristics.

For each calculated dosage, the corresponding steady-state Cmin was calculated using Equation (6), with the true simulated parameter values and the calculated dose and dosing interval. Percentages of *Cmin* within the target interval, below the lower bound and above the upper bound, were calculated for each dosing method.

### 2.6. Evaluation of TID and Other Dosing Methods Based on Data from Real Patients

Individual estimates of imatinib CL and V, as well as covariate values were available from two groups of 65 and 20 patients, all with GIST. These patients’ characteristics as well as the Bayesian method used to estimate imatinib PK parameters and ethics committee approval have been described in a previous publication [27]. This study was approved by the ethics committee of the Lausanne Faculty of Medicine (approval number 31/09, and informed written consent was obtained from the participants.

For each patient, at least one measured concentration of imatinib was available from routine TDM. In 20 patients, a pair of concentrations measured over the same dosing interval was available on at least one occasion. Individual values of *CL* and *Vd* were estimated for each patient on each occasion using a (MAP) method based on the model from Widmer et al. [26]. Ka was fixed at the population value of 0.609 in all patients. When multiple CL and Vd values were available from several occasions in the same patient, the mean estimate was used in subsequent calculations.

We used these MAP estimates as the “true value” to assess the ability of the various imatinib dosing methods to achieve a target Cmin interval. The target interval was fixed at 1100–2000 ng/mL in this case, as Cmin ≥ 1100 ng/mL is the target value recommended in patients with GIST [30].

A procedure similar to that previously described for simulated patients was carried out. For the TID and TCD approaches, several dosage regimens of imatinib were calculated, based on the patients’ individual covariates (body weight, age, gender and disease, here GIST), as described for simulated patients. Fixed-dose regimens were also evaluated.

We then computed the predicted steady-state Cmin corresponding to each calculated dosage, and the percentage of Cmin within and outside the desired target interval, as described previously, using Equation (6) and the individual MAP estimates of *CL* and *Vd*.

### 2.7. Model-Based Dosing Recommendations

We used the TID–OD_int_ method to derive dosage recommendations according to age, body weight, sex and disease (CML or GIST). We set the Cmin target interval to 1000–2000 ng/mL to accommodate both CML and GIST. We only considered once daily and twice daily dosage, with daily dose ranging from 100 to 800 mg per day. The selected dosage was that with the lowest dose and the largest dosing interval which maximized the a priori probability to achieve the target interval, as less frequent dosing regimens are desirable for patient’s adherence [34].

### 2.8. Software Tools

The model discretization, random sampling for the 200-subject simulations and the calculation of the optimal dosage based on the TID approach were performed using Matlab software (version 2018b, Mathworks, Natick, MA, USA).

## 3. Results

### 3.1. Imatinib Dosing and Target Attainment in Simulated Patients

Table 1 shows the performance of the various dosing methods in achieving the target intervals of 1000–2000 ng/mL in all simulated patients (*n* = 800). Figure 4 shows the distributions of Cmin achieved with the various dosing methods evaluated.

The TID-OD_int_, TID-OD_L5%_ as well as the two TCD-based dosages (q24h and q12h) provided the highest percentages of target attainment, around 65%. Regimens based on those methods were associated with lower variability in Cmin and a better distribution of values within the target interval, compared with fixed-dose regimens. Those fixed-dose regimens provided poor rates of target attainment, especially the 400 mg/24 h regimen showing only 29% of Cmin within the target range. This regimen was also associated with a high proportion of underexposure (68.2%). The fixed-dose regimen with 600 mg/24 h or 300 mg/12 h of imatinib provided better achievement of the target interval (>50%) but were associated with large proportions of underexposure and overexposure, respectively. This confirms that any fixed-dose regimen is unlikely to be optimal for achieving an imatinib target concentration in all patients, because of PK variability.

As predicted by theory, the TID-OD_Lmin_ and TID-OD_L5%_ methods were effective in minimizing underexposure, but this logically resulted in higher rates of overexposure, especially for the OD_Lmin_ method. Compared with the latter method, OD_L5%_ appears as a more reasonable compromise to minimize underexposure while limiting overexposure. Of note, the observed frequency of underexposure with this method was higher than expected (10.9% versus 5%). This discrepancy is likely due to a proportion of simulated patients with high imatinib CL and V values greater than expected based on the discrete prior distribution.

There were some differences in the results between the four patient subgroups. While the percentages of target interval achievement were relatively consistent across gender and disease groups for model-based dosages, they markedly varied for the fixed-dose regimens. For example, for the ODint, TCD q24h, 400 mg q24h and 300 mg q12h dosages, the respective proportions of Cmin within 1000–2000 ng/mL were as follows: 65%, 66%, 31% and 59% in women with CML; 68%, 65%, 49% and 41% in women with GIST; 60%, 55%, 14% and 61% in men with CML; 72%, 70%, 22% and 67% in men with GIST. The 400 mg q24h dosage was associated with higher rates of underexposure in men than in women, because of increased clearance predicted by the model.

Table 2 presents the characteristics of dosages based in the TID method. While twice daily dosing was the most frequent schedule for the methods minimizing underexposure, thrice daily and once daily dosing were found as the optimal schedule in a significant proportion of patients with each method. As expected, minimizing underexposure with the OD_Lmin_ and OD_L5%_ methods required larger mean doses than that maximizing attainment of the target interval (OD_int_).

### 3.2. Performance of Imatinib Dosing Methods Based on Real Patients’ Data

Table 3 shows the demographic and PK data of the 85 patients with GIST. Table 4 summarizes the simulated imatinib Cmin and attainment of the target interval for each dosing method in those 85 patients. The model-based TID-OD_int_ and TCD q24h methods performed best in terms of target interval attainment, but the fixed regimen with 300 mg/12 h also performed remarkably well in this setting and was associated with a very low proportion of Cmin < 1100 ng/mL.

Table 5 provides dosage recommendations based on the TID-OD_int_ to achieve imatinib Cmin within the 1000–2000 ng/mL interval. As predicted by the population PK model, the youngest and heaviest patients would require the larger daily dose and twice-daily administration to achieve the target Cmin, while lower doses administered once daily would be sufficient for older patients with low body weight.

## 4. Discussion

The suboptimal exposure associated with dosage recommendations approved at the time of commercialization of anticancer agents is increasingly recognized and justifies efforts towards dosage optimization after drug approval [35].

Because of increasing evidence of concentration-effect relationships, achieving therapeutic concentration of tyrosine kinases inhibitors has become a relevant objective in clinical practice [36,37]. As imatinib pharmacokinetic variability is large, with a 20-fold variability in steady-state trough concentrations in patients receiving the same dosage [16], imatinib dose is poorly predictive of drug exposure. The standard initial dose of imatinib of 400 mg has not been designed to achieve a target Cmin ≥ 1000 ng/mL. Larson et al. reported steady-state Cmin 25th, 50th and 75th percentile values of 647, 879 and 1170 ng/mL in 351 CML patients who received 400 mg of imatinib [16]. In another study in 108 patients with GIST, with 80% of them receiving 400 mg/24 h, 44.4% of patients had their first imatinib Cmin at steady-state < 1000 ng/mL [38]. This means that many patients under this dosage do not achieve the target concentration in both conditions and that precision dosing of imatinib is desirable.

The use of PK models is the state-of-the-art approach for designing drug dosage regimen targeting a specific exposure [39,40]. The traditional, target concentration dosing (TCD), approach for model-based initial drug dosing consists in using a single set of PK parameters, usually based on a population PK study, to calculate the dose necessary to achieve a single target value of the exposure index of interest (e.g., trough concentration, maximal concentration or area under the curve) within the therapeutic interval. Recently, we have developed an alternative model-based dosing approach that specifically optimizes the probability to achieve a concentration or exposure interval. We have previously evaluated this target interval dosing (TID) method for busulfan dosing in children [28], and this approach outperformed other model-based approaches in terms of target attainment.

In this study, we applied the TID method for imatinib dosing and compared its performance with that of other dosing approaches in simulated patients and using real patients’ data. When targeting the imatinib Cmin interval of 1000–2000 ng/mL, the TID approach was effective in maximizing the rate of target attainment. However, in this example, it did not perform better than the traditional model-based approach targeting the mid-interval (TCD). This result may not be generalizable as the ability to achieve a target interval can depend on several features including the variability of PK parameters and the width of the target interval. However, the TCD method should be effective when the condition of a relatively symmetrical distribution of the exposure around the median is fulfilled, while the TID approach may be superior in case of heavy-tailed distribution of the exposure index [28]. The latter also has the unique characteristic of accommodating single bound intervals, and so can minimize under- or overexposure. In this application to imatinib, the TCI-OD_Lmin_ and -OD_L5%_ methods consistently minimized the proportion of Cmin < 1000 mg/L. However, these methods were associated with the highest dosage requirements and higher rates of Cmin > 2000 ng/mL, which may raise safety concerns. This illustrates the limitations of any dosing method based on population data only and the need to perform TDM for individual control of drug exposure during treatment [27,41].

Imatinib TDM has been officially recommended by the International Association of Therapeutic Drug Monitoring and Clinical Toxicology for patients with CML and GIST in 2021 [24]. The dosages based on the TID approach presented in Table 5 could be used when no TDM is possible, or before TDM can be performed, to optimize the achievement of the Cmin target interval of 1000 to 2000 ng/mL.

Our study results also illustrate the limitations of the “one size fits all” approach for imatinib dosing. The standard 400 mg/24 h dosage was associated with very low rates of simulated Cmin ≥ 1000 ng/mL, especially in male patients, which confirms the results of several observational studies [16,21]. Based on our simulations, twice-daily dosing with 200 mg or 300 mg of imatinib every 12 h would enhance the achievement of Cmin target. However, a large variability in exposure and significant proportions of under or overexposure would still occur because fixed-dose approaches do not consider PK variability and are not goal-oriented. Only model-based approaches can consistently reduce the variability and control the drug exposure optimally in all patients’ group identified by population PK. Cost savings may also be expected by optimizing the dosage and hence the efficacy of imatinib, now that generic copies are available, while second-line treatments remain expensive [25].

This study is based on several assumptions and has a number of limitations that should be acknowledged. First, all model-based dosages (with both the TID and TCD approaches) and dosing recommendations were based on the model from Widmer et al. [26]. So, it is assumed that this model adequately describes imatinib concentration, at least Cmin. While other population PK models of imatinib have been published [42], this model has been validated for Bayesian analysis of imatinib concentration in routine TDM conditions [27]. In addition, the concentrations simulated with this model are in close agreement with real data. The 25th, 50th and 75th percentiles of Cmin values simulated in 800 patients for the 400 mg/24 h dosage were 611, 804 and 1099 ng/mL and compared well with the values from 351 patients in the study from Larson et al. presented above [16]. However, as true parameter values of both simulated and real patients were generated using the Widmer model, these conditions may favor the performance of the model-based approaches. This is especially true for the evaluation based on real patient data, as the individual Bayesian estimates of PK parameters are likely to shrink towards the typical values and so favor the TCD approach. A prospective clinical evaluation of such model-based approaches is necessary to confirm their interest in controlling imatinib Cmin. The dosages suggested based on this modeling approach (Table 5) were defined for specific age (20 to 90 years) and weight (40 to 110 kg) ranges and cannot be extrapolated out of those ranges. They were based on the TID-OD_int_ approach that should minimize both under- and overexposure. The other TID methods (OD_Lmin_ and OD_L5%_) could lead to different dosages. 

We did not directly consider safety in the dosing. The upper bound of the target interval was arbitrarily fixed at 2000 ng/mL, i.e., twice the lower bound value. Although rates of adverse reactions to imatinib such as fluid retention, rash and anemia were found to rise with increasing Cmin [16,23,42], there is currently no clear recommendation regarding the maximum Cmin that should not be exceeded [24]. In their meta-analysis, Garcia-Ferrer et al. discussed this upper limit of imatinib Cmin, mentioning both 3000 ng/mL and 1500 ng/mL, but recognized the lack of evidence. The upper bound used in our study was in between and appears reasonable. As the TID-OD_int_ approach is based on targeting a Cmin interval, not only exceeding the efficacy threshold, we believe it is relevant to prevent high exposure that might be more toxic. Finally, our dosing approach does not handle the influence of other covariates such as drug–drug interaction. Cytochrome P450 3A4-mediated drug interactions may significantly alter imatinib exposure [43]. This could be considered in further development of the approach. TDM is probably the best way to deal with this risk at the individual level.

## 5. Conclusions

This study confirms that fixed-dose regimens of imatinib currently recommended are associated with low Cmin in most patients with CML or GIST. Precision dosing of imatinib based on PK models can improve the achievement of target imatinib Cmin in patients treated for CML or GIST. Our target interval dosing approach has the ability to optimize the achievement of the imatinib target Cmin interval, as well as minimizing underexposure. Dosing recommendations based on this approach are suggested. While model-based dosing can improve initial dosing, TDM remains necessary to control the individual exposure and adjust the dosage during therapy. The combination of initial model-based dosing and subsequent dose individualization based on TDM are sound basis for precision medicine with imatinib therapy and other drugs with exposure–response relationships in oncology.

## Figures and Tables

**Figure 1 pharmaceutics-15-01081-f001:**
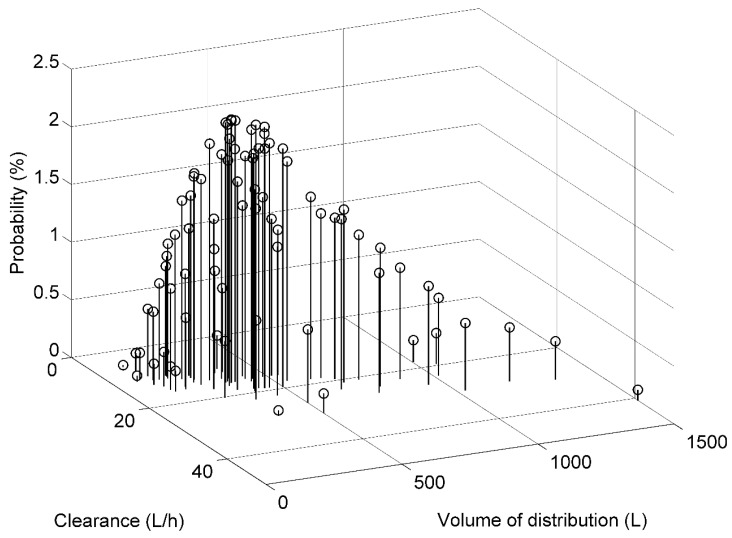
**Discrete prior distribution of imatinib clearance and volume of distribution in a typical patient.** The patient is a male with CML, his age and body weight are 50 years and 70 kg, respectively.

**Figure 2 pharmaceutics-15-01081-f002:**
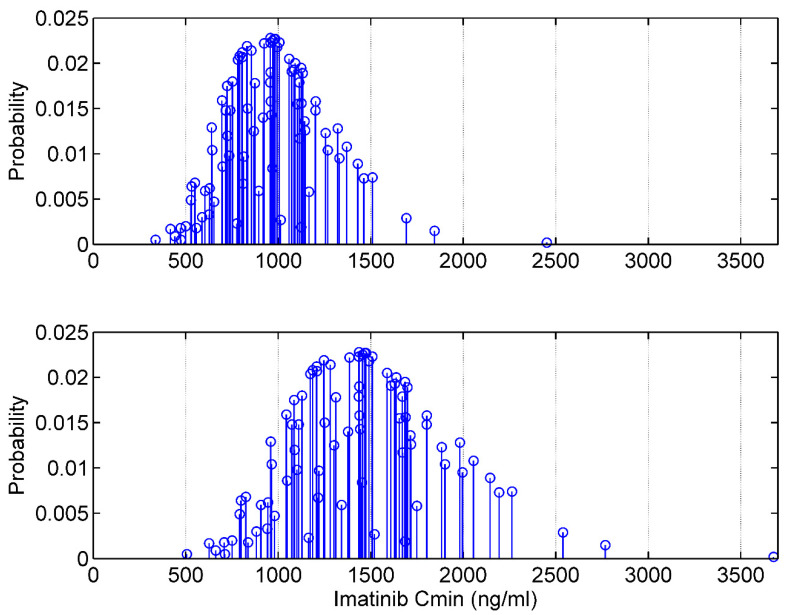
**A priori discrete distribution of imatinib trough concentration at the steady state in a virtual patient.** Upper panel, 400 mg/24 h; lower panel, 600 mg/24 h. The patient is a 60-year-old woman, her weight is 60 kg and she has CML.

**Figure 3 pharmaceutics-15-01081-f003:**
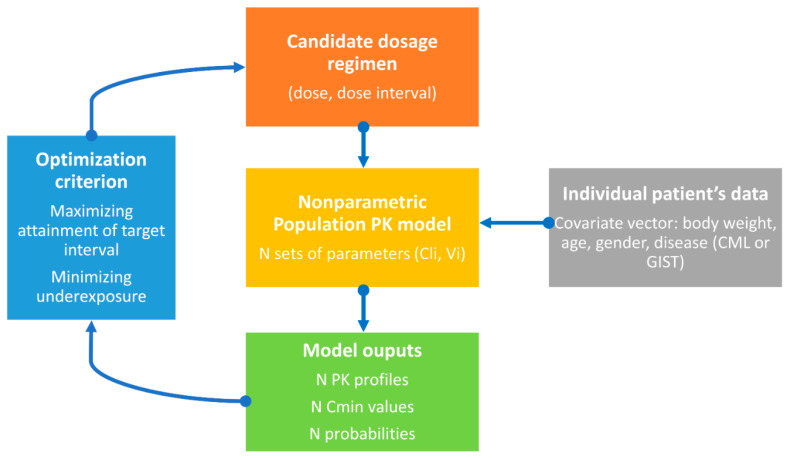
Graphical abstract of the target interval dosing (TID) approach for imatinib.

**Figure 4 pharmaceutics-15-01081-f004:**
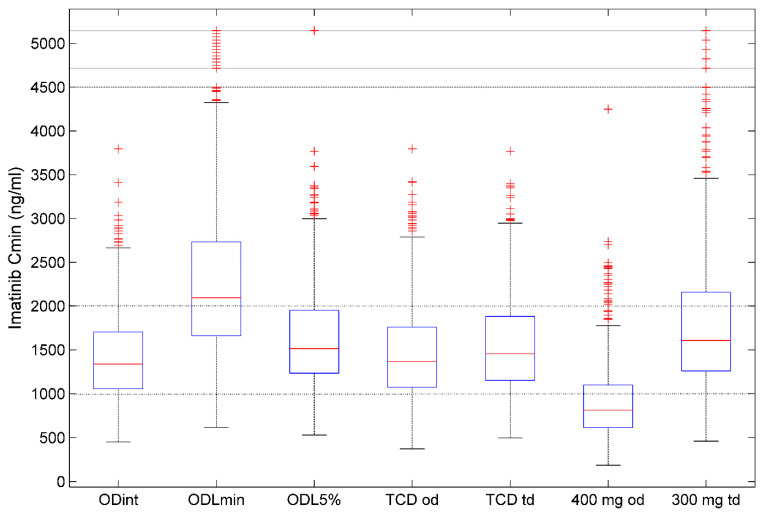
**Boxplot of imatinib trough concentrations achieved in simulated patients with various dosing methods targeting the interval 1000–2000 ng/mL.** The box edges represent the 25th and 75th percentiles of Cmin distributions. The central mark is the median. The length of whiskers is equal to 1.5 times the interquartile range. The plus signs are outlier values. For ease of graphical display, Cmin values greater than 4500 ng/mL are all compressed in the upper area between the two grey solid lines. Of note, the proportions of Cmin values within and outside the target interval are slightly different from those shown in Table 1 because the data represented graphically are restricted to the 791 Cmin values that could be computed with all methods. Abbreviations: Cmin, trough concentration; od, once daily administration (every 24 h); td, twice daily administration (every 12 h); TCD, target concentration dosing.

**Table 1 pharmaceutics-15-01081-t001:** Imatinib Cmin and attainment of target interval (1000–2000 ng/mL) for the various dosing methods in all simulated patients.

	TID-OD_int_	TID-OD_Lmin_	TID-OD_L5%_	TCD q24h	TCD q12h	400 mg/24 h	600 mg/24 h	200 mg/12 h	300 mg/12 g
Number of predictions	800	800	791 ^a^	800	800	800	800	800	800
Mean imatinib dose per dosing interval (mg)	NA	NA	NA	683 (131)	278 (68)	400	600	200	300
Mean Cmin (ng/mL)	1420	2252	1624	1458	1553	902	1353	1182	1773
Coefficient of variation of Cmin (%)	35%	39%	35%	37%	36%	48%	48%	44%	44%
Median Cmin (ng/mL)	1340	2087	1515	1365	1454	804	1205	1069	1603
5th and 95th percentiles of Cmin (ng/mL)	755–2325	1118–3959	850–2727	739–2433	807–2626	398–1694	597–2541	559–2164	839–3245
Cmin within target interval (%)	66.0%	43.1%	66.2%	64.5%	64.6%	29%	54.7%	49.3%	56.6%
Cmin < 1000 ng/mL (%)	20.2%	2.8%	10.9%	18.9%	15.6%	68.2%	32%	43.2%	13.4%
Cmin > 2000 ng/mL (%)	13.8%	54.1%	22.9%	16.6%	19.8%	2.8%	13.3%	7.3%	30%

^a^ The dosage calculation with OD_L5%_ method was impossible in 9 simulated patients because the calculated probability of underexposure was > 5%. Abbreviations: NA, not applicable.

**Table 2 pharmaceutics-15-01081-t002:** Characteristics of dosing regimens based on the TID approach in simulated patients.

		OD_int_	OD_Lmin_	OD_L5%_
Dosing schedule ^a^	q8h	12%	2.5%	16.2%
	q12h	21%	95.3%	67.5%
	q24h	67%	2.2%	16.3%
Mean dose per dosing interval	q8h	180 mg	200 mg	188 mg
	q12h	266 mg	394 mg	299 mg
	q24h	639 mg	611 mg	581 mg

^a^ The percentages indicate the proportion of q8h, q12h, and q24h schedule for each method.

**Table 3 pharmaceutics-15-01081-t003:** Characteristics of the 85 patients with GIST.

Variable	Value
Number of females/males	44/41
Age (years)	62 ± 13 (23–85)
Body weight (kg)	71.2 ± 13.2 (48–100)
Imatinib clearance (L/h)	12.9 ± 3.7 (2.6–24.4)
Imatinib volume of distribution (L)	376 ± 146 (51–717)

**Table 4 pharmaceutics-15-01081-t004:** Predicted imatinib C_min_ and attainment of target interval (1100–2000 ng/mL) in 85 patients with GIST.

	TID-OD_int_	TID-OD_Lmin_	TID-OD_L5%_	TCD q24h	TCD q12h	400 mg/24 h	600 mg/24 h	200 mg/12 h	300 mg/12 h
Mean Cmin (ng/mL)	1462	2369	1754	1514	1543	945	1417	1216	1824
Coefficient of variation of Cmin (%)	39%	44%	49%	37%	39%	40%	40%	44%	44%
Median Cmin (ng/mL)	1303	2208	1547	1413	1344	886	1329	1130	1695
5th and 95th percentiles of Cmin (ng/mL)	980–2144	1475–3744	1040–2944	1001–2109	966–2390	567–1488	851–2233	753–1896	1130–2845
Cmin within target interval (%)	75.3%	36.5%	64.7%	80.0%	72.9%	16.5%	65.9%	51.8%	72.9%
Cmin < 1100 ng/mL (%)	16.5%	0%	9.4%	11.8%	14.1%	82.3%	24.7%	45.9%	3.5%
Cmin > 2000 ng/mL (%)	8.2%	63.5%	25.9%	8.2%	13.0%	1.2%	9.4%	2.3%	23.6%

**Table 5 pharmaceutics-15-01081-t005:** Imatinib dosage recommendations based on the TID approach according to age, body weight, sex and disease. The suggested dosage regimens (dose in mg/dosing interval in hours) are those calculated with the TID-ODint approach to maximize the probability to achieve a Cmin target interval of 1000–2000 ng/mL at the steady-state. The color code is as follows.

	Age (Years)															
Body Weight (kg)	20		30		40		50		60		70		80		90	
40	400/12	300/12	300/12	800/24	300/12	700/24	700/24	600/24	600/24	200/12	600/24	400/24	200/12	400/24	400/24	300/24
	300/12	700/24	800/24	600/24	700/24	200/12	600/24	500/24	500/24	400/24	500/24	300/24	400/24	100/12	300/24	200/24
50	400/12	300/12	400/12	300/12	300/12	700/24	800/24	600/24	700/24	600/24	600/24	500/24	500/24	400/24	200/12	300/24
	300/12	800/24	800/24	700/24	800/24	600/24	700/24	500/24	600/24	400/24	500/24	400/24	400/24	300/24	400/24	100/12
60	400/12	400/12	400/12	300/12	300/12	800/24	800/24	700/24	800/24	600/24	700/24	500/24	600/24	400/24	500/24	400/24
	400/12	300/12	300/12	800/24	300/12	700/24	700/24	200/12	600/24	500/24	600/24	400/24	200/12	300/24	400/24	300/24
70	400/12	400/12	400/12	300/12	400/12	300/12	300/12	800/24	300/12	700/24	700/24	600/24	600/24	200/12	600/24	400/24
	400/12	300/12	400/12	300/12	300/12	700/24	800/24	600/24	700/24	200/12	600/24	500/24	500/24	400/24	500/24	300/24
80	400/12	400/12	400/12	400/12	400/12	300/12	400/12	300/12	300/12	700/24	800/24	600/24	700/24	600/24	600/24	500/24
	400/12	400/12	400/12	300/12	300/12	800/24	800/24	700/24	800/24	600/24	700/24	500/24	600/24	400/24	500/24	400/24
90	400/12	400/12	400/12	400/12	400/12	400/12	400/12	300/12	300/12	800/24	800/24	700/24	800/24	600/24	700/24	500/24
	400/12	400/12	400/12	300/12	400/12	300/12	300/12	800/24	300/12	700/24	700/24	200/12	600/24	500/24	600/24	400/24
100	400/12	400/12	400/12	400/12	400/12	400/12	400/12	300/12	400/12	300/12	300/12	800/24	300/12	700/24	700/24	600/24
	400/12	400/12	400/12	400/12	400/12	300/12	400/12	300/12	300/12	700/24	800/24	600/24	700/24	200/12	600/24	500/24
110	400/12	400/12	400/12	400/12	400/12	400/12	400/12	400/12	400/12	300/12	400/12	300/12	300/12	700/24	800/24	600/24
	400/12	400/12	400/12	400/12	400/12	400/12	400/12	300/12	300/12	800/24	800/24	700/24	700/24	600/24	700/24	500/24
	Man with CML			Woman with CML												
	Man with GIST			Woman with GIST												

## Data Availability

Data are available upon request.

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
