# Peer review of "From Personalized to Precision Medicine in Oncology: A Model-Based Dosing Approach to Optimize Achievement of Imatinib Target Exposure"

_pharmaceutics, 2023, doi:10.3390/pharmaceutics15041081_

Round 1

Reviewer 1 Report

This is an interesting simulation study result using a well-established pharmacokinetic model, and the reliability of the result was increased by verifying the performance of the model using covariates of real patients. Since the content has been written in detail and the results presented in detail, it is believed to be sufficient for the reader to understand. 

1.  The parameter values presented in Eq 1. and Eq 2. are slightly different from the values in Appendix I of the reference (Br J Clin Pharmacol 2006 Jul; 62(1): 97–112) presented. Any further explanation on how those values were derived would be more helpful.

2. The ranges of age and weight in the data used for the construction of the pharmacokinetic model are presented in the references cited above. In addition, the predictive performance of this model appears to be validated over a wider age range (reference No. 27 / Clinical pharmacokinetics. 2012;51(3):187-201). Considering the range used in the simulation included in this study and the distribution of covariates in actual patients, please suggest the range of weight and age to which the results of this study can be applied in discussion.

3. It is described in Line 347 that the dosage recommendation using TID-ODint was made. It would be helpful if you could describe more clearly whether this was defined in advance or whether the most appropriate method was selected through simulation results.

4. TID-based methods did not yield better results than TCD q24h. Perhaps this is because when the dose was recommended based on a single value (target Cmin, ), the size of variation that could occur in the group (eg, 95% confidence interval) was not greater than the width of the target Cmin interval. It would be helpful for readers' understanding if you mention once more the size of the confirmed dispersion along with the symmetry of the distribution.

Author Response

1/ The parameter values presented in Eq 1. and Eq 2. are slightly different from the values in Appendix I of the reference (Br J Clin Pharmacol 2006 Jul; 62(1): 97–112) presented. Any further explanation on how those values were derived would be more helpful.

R. Appendix I of the paper from Widmer et al. (Br J Clin Pharmacol 2006 Jul; 62(1): 97–112) describes the covariate model building. The parameter values indicated in this appendix were not the final parameter values. The final parameter values are presented in Table 2 of this paper, and we used this value in the present study.

2/ The ranges of age and weight in the data used for the construction of the pharmacokinetic model are presented in the references cited above. In addition, the predictive performance of this model appears to be validated over a wider age range (reference No. 27 / Clinical pharmacokinetics. 2012;51(3):187-201). Considering the range used in the simulation included in this study and the distribution of covariates in actual patients, please suggest the range of weight and age to which the results of this study can be applied in discussion.

R. Indeed, we used age and weight ranges based on the validation study from Gotta et al. (reference #27), which was larger than the original study from Widmer et al. This is indicated in the manuscript, page 9: “These values were representative of individual data from 85 patients with GIST treated with imatinib [27].“. A comment has been added in the discussion to indicate that results are valid within those ranges.

3/ It is described in Line 347 that the dosage recommendation using TID-ODint was made. It would be helpful if you could describe more clearly whether this was defined in advance or whether the most appropriate method was selected through simulation results.

R. The TID-ODint method showed its ability to minimize both under- and overexposure (see Table 1), which is a natural objective. The other TID methods are designed to minimize underexposure, but we thought that this goal was more arbitrary. There is no formal agreement about which percentage of underexposure is acceptable. A comment has been added in the discussion to recognize that another approach could lead to different dosage recommendations.

4/ TID-based methods did not yield better results than TCD q24h. Perhaps this is because when the dose was recommended based on a single value (target Cmin, ), the size of variation that could occur in the group (eg, 95% confidence interval) was not greater than the width of the target Cmin interval. It would be helpful for readers' understanding if you mention once more the size of the confirmed dispersion along with the symmetry of the distribution.

R. Thank you for this interesting comment. The size of the dispersion is well described in Figure 4 as well as Tables 1 and 4. It is true that the interquartile range of predicted Cmin is included with the target Cmin interval, but the 90% prediction interval stretches well outside. We agree that the performance of the model-based approach depends on several model-based features (CV% of PK parameters estimated from the population PK study) and arbitrary criterion such as the width of the target interval. A comment has been added in the discussion.

Reviewer 2 Report

A well-written manuscript. Authors seek to achieve better dosing for cancer patients receiving imatinib a priori, for cases where therapeutic drug monitoring (TDM) may not be feasible.

Quite often safety is related to Cmax (peak exposure) or AUC and not Cmin; authors should simulate, discuss and disclose the probability of breaching an upper limit of safety with the proposed dosing table (Table 5). 

More frequent and shorter dosing interval will always improve Cmin target attainment since Cmin will approach Cavg as tau decreases toward continuous. Could the authors then clarify how the decision is made for so many once daily doses in the recommendations in Table 5; this must be factoring in patient convenience or weighted to favor once daily doses? Why bother with all of the modeling if you can just dose all patients with 300-400 mg twice daily based on body weight and tolerability, obviating the need for the M&S paper? Please discuss.

Caption for Figure 3 is out of place on page 7.

Author Response

A well-written manuscript. Authors seek to achieve better dosing for cancer patients receiving imatinib a priori, for cases where therapeutic drug monitoring (TDM) may not be feasible.

Quite often safety is related to Cmax (peak exposure) or AUC and not Cmin; authors should simulate, discuss and disclose the probability of breaching an upper limit of safety with the proposed dosing table (Table 5).

R. As explained in the article introduction, there is no sound evidence about exposure-toxicity relationships for imatinib. However, it should be acknowledged that most studies only performed Cmin measurement. AUC or Cmax might better predict the risk of adverse event, but there are no data supporting this assumption, and no safety limits have been defined for Cmax or AUC. This is why this was not considered in the present study.

More frequent and shorter dosing interval will always improve Cmin target attainment since Cmin will approach Cavg as tau decreases toward continuous. Could the authors then clarify how the decision is made for so many once daily doses in the recommendations in Table 5; this must be factoring in patient convenience or weighted to favor once daily doses? Why bother with all of the modeling if you can just dose all patients with 300-400 mg twice daily based on body weight and tolerability, obviating the need for the M&S paper? Please discuss.

R. This is correct, more frequent dose intakes should improve Cmin target attainment. However, it has been shown that more frequent dose intakes are associated with poorer drug adherence (see https://doi.org/10.1016/S0149-2918(01)80109-0). This is why we selected dosages with the lowest dose and the largest dosing interval when several regimens maximized probability to achieve the target interval, as explained in section 2.7. A comment has been added in section 2.7 to justify this choice.

In addition, a fixed-dose regimen such as 300-400 mg twice daily will result in higher proportions of Cmin > 2000 ng/L (see Figure 4 and Table 1). This is not precision dosing.

Caption for Figure 3 is out of place on page 7.

R. Correction done, thank you.

Reviewer 3 Report

                The authors present an interesting study summary in which they used both traditional and two model-based methods for optimizing imatinib dosing for patients with chronic myeloid leukemia or gastrointestinal stromal tumor.  The focus on this dose optimization was to ensure an adequate Cmin for patient response based on previous studies concerning imatinib exposure and efficacy. Overall, this is a very interesting modeling exercise with good clinical implications. Below are some major and minor comments for the authors to consider.

Major Comments:

-Authors seem to discount the risk of adverse events (AEs) related to imatinib exposure, particularly higher exposure. In fact stating in the introduction that “Although some adverse reactions were more frequent in patients with the highest Cmin in the study from Larson and colleagues [16], data supporting an upper bound for imatinib concentration are limited [20].” While the concentration threshold for AEs is not explicitly elucidated, it is well known that tyrosine kinase inhibitors, including imatinib, commonly cause rashes, skin lesions, and edema at fairly high rates. (https://onlinelibrary.wiley.com/doi/full/10.1046/j.1365-2141.2003.04151_4.x)  Studies have proposed that this is due to the pharmacological action of the drug, thus higher concentrations will likely increase the already relatively high rate of such AEs. (https://www.nature.com/articles/6600893). Thus, this reviewer believes focusing solely on increasing Cmin without acknowledging the risk of increasing AE rates is a major limitation of this work and should be acknowledged throughout the manuscript (not just the last lines at the end of the discussion), especially since they are making clinical suggestions.

-It is known that imatinib is susceptible to CYP3A4 drug-drug interactions (both as victim and perpetrator), which will of course impact its PK in conditions where comedications also acting on CYP3A4 are ingested. Since these are sick patients it is likely that they will be taking other drugs and potentially at risk for such DDIs. Thus, it is suggested that authors address this clinically-relevant scenario in their work either by acknowledging this as a barrier to clinical implementation of their dosing projections or demonstrating that the proposed models can account for these scenarios.

Minor Comments:

-It is unclear what the sentence on lines 190-191 are referring to. It should either be removed or clarified. “The patient is a male with CML, his age and body weight are 50 years and 70 kg, respectively.”

-“weighting” should be changed to “weighing” in the sentence on line 195.

-“his” should be changed to “her” on line 196 as the previous sentence indicates that the example is of a 60 year-old woman.

Author Response

The authors present an interesting study summary in which they used both traditional and two model-based methods for optimizing imatinib dosing for patients with chronic myeloid leukemia or gastrointestinal stromal tumor.  The focus on this dose optimization was to ensure an adequate Cmin for patient response based on previous studies concerning imatinib exposure and efficacy. Overall, this is a very interesting modeling exercise with good clinical implications. Below are some major and minor comments for the authors to consider.

 Major Comments:

-Authors seem to discount the risk of adverse events (AEs) related to imatinib exposure, particularly higher exposure. In fact stating in the introduction that “Although some adverse reactions were more frequent in patients with the highest Cmin in the study from Larson and colleagues [16], data supporting an upper bound for imatinib concentration are limited [20].” While the concentration threshold for AEs is not explicitly elucidated, it is well known that tyrosine kinase inhibitors, including imatinib, commonly cause rashes, skin lesions, and edema at fairly high rates. (https://onlinelibrary.wiley.com/doi/full/10.1046/j.1365-2141.2003.04151_4.x)  Studies have proposed that this is due to the pharmacological action of the drug, thus higher concentrations will likely increase the already relatively high rate of such AEs. (https://www.nature.com/articles/6600893). Thus, this reviewer believes focusing solely on increasing Cmin without acknowledging the risk of increasing AE rates is a major limitation of this work and should be acknowledged throughout the manuscript (not just the last lines at the end of the discussion), especially since they are making clinical suggestions.

R. This is correct of course, imatinib and other tyrosine-kinase inhibitors may be responsible for serious adverse reactions. But, since no clear exposure-toxicity relationship and no specific concentration target has been described, this risk cannot be directly considered in our dosing approach. However, as our dosing approach is based on targeting a Cmin interval (1000 to 2000 ng/ml), not only exceeding the efficacy threshold, we believe it is relevant to prevent high exposure that might be toxic. A comment has been added in the discussion.

-It is known that imatinib is susceptible to CYP3A4 drug-drug interactions (both as victim and perpetrator), which will of course impact its PK in conditions where comedications also acting on CYP3A4 are ingested. Since these are sick patients it is likely that they will be taking other drugs and potentially at risk for such DDIs. Thus, it is suggested that authors address this clinically-relevant scenario in their work either by acknowledging this as a barrier to clinical implementation of their dosing projections or demonstrating that the proposed models can account for these scenarios.

R. We agree, our current approach does not handle the influence of drug-drug interaction on imatinib exposure. This could be possible by using another population model that would include and quantify this effect, but it is quite complex because the effect of DDI depends on the potency of the inducer/inhibitor, which is highly variable. We believe this is out of the scope of our work, but should be considered in further development. Therapeutic drug monitoring is probably the best way to deal with this issue at the individual level. This limitation has been added in the discussion.

Minor Comments:

-It is unclear what the sentence on lines 190-191 are referring to. It should either be removed or clarified. “The patient is a male with CML, his age and body weight are 50 years and 70 kg, respectively.”

R. This sentence should be included in the figure caption above. Correction done.

-“weighting” should be changed to “weighing” in the sentence on line 195.

R. Correction done, thank you.

-“his” should be changed to “her” on line 196 as the previous sentence indicates that the example is of a 60 year-old woman.

R. Correction done, thank you.

Reviewer 4 Report

Very interesting work, well written publication and applicable in the clinical setting. Methods are well described and the limitations of the work are addressed in the discussion. No further changes are suggested.

Author Response

Very interesting work, well written publication and applicable in the clinical setting. Methods are well described and the limitations of the work are addressed in the discussion. No further changes are suggested.

R. Thanks a lot !